# Oxolane Ammonium Salts (Muscarine-Like)—Synthesis and Microbiological Activity

**DOI:** 10.3390/ijms25042368

**Published:** 2024-02-17

**Authors:** Patrycja Bogdanowicz, Janusz Madaj, Piotr Szweda, Artur Sikorski, Justyna Samaszko-Fiertek, Barbara Dmochowska

**Affiliations:** 1Faculty of Chemistry, University of Gdansk, Wita Stwosza 63, 80-308 Gdansk, Poland; patrycja.bogdanowicz99@wp.pl (P.B.); janusz.madaj@ug.edu.pl (J.M.); artur.sikorski@ug.edu.pl (A.S.); j.samaszko-fiertek@ug.edu.pl (J.S.-F.); 2Department of Pharmaceutical Technology and Biochemistry, Gdansk University of Technology, Gabriela Narutowicza Street 11/12, 80-233 Gdansk, Poland; piotr.szweda@pg.edu.pl

**Keywords:** oxolane ring, muscarine, 2-deoxy-D-ribose, ammonium salts, microbiological tests

## Abstract

Commercially available 2-deoxy-D-ribose was used to synthesize the appropriate oxolane derivative—(2*R*,3*S*)-2-(hydroxymethyl)oxolan-3-ol—by reduction and dehydration/cyclization in an acidic aqueous solution. Its monotosyl derivative, as a result of the quaternization reaction, allowed us to obtain eight new muscarine-type derivatives containing a quaternary nitrogen atom and a hydroxyl group linked to the oxolane ring. Their structure was fully confirmed by the results of NMR, MS and IR analyses. The crystal structure of the pyridinium derivative showed a high similarity of the conformation of the oxolane ring to previously published crystal structures of muscarine. Two reference strains of Gram-negative bacteria (*Escherichia coli* ATCC 25922 and *Pseudomonas aeruginosa* ATCC 27853), two reference strains of Gram-positive staphylococci (*Staphylococcus aureus* ATCC 25923 and *Staphylococcus aureus* ATCC 29213) and four reference strains of pathogenic yeasts of the genus *Candida* spp. (*Candida albicans* SC5314, *Candida glabrata* DSM 11226, *Candida krusei* DSM 6128 and *Candida parapsilosis* DSM 5784) were selected for the evaluation of the antimicrobial potential of the synthesized compounds. The derivative containing the longest (decyl) chain attached to the quaternary nitrogen atom turned out to be the most active.

## 1. Introduction

Oxolane (tetrahydrofuran) is a simple heterocyclic compound. Its five-membered ring contains four carbon atoms and a heterocyclic oxygen atom. A number of methods for the synthesis of both tetrahydrofuran [1] and its derivatives [2,3,4,5,6] have been described in the literature. Despite having such a simple structure, it is a fragment of the structure of many natural and synthetic compounds with a wide spectrum of biological activity. These include furanose forms of simple sugars, especially D-ribose and 2-deoxy-D-ribose, as components of nucleosides, which are components of DNA and RNA nucleic acids. The literature also describes the synthesis of nucleoside analogues, which are also characterized by a number of interesting biological activities [7,8,9,10,11]. Derivatives containing an oxolane ring are common in nature, including in marine organisms. Such derivatives include lipid alcohols [12,13], fatty acids [14] and terpenes [15,16,17,18,19].

A very interesting derivative containing an oxolane ring is muscarine (Figure 1).

This simple chemical compound owes its name to the *Amanita muscaria* mushroom, from which it was first isolated by Schmiedeberg and Koppe [20]. Its structure was confirmed by Chan and Li, who performed the first chemical synthesis of this compound [21]. Ultimately, the structure was fully confirmed by the crystallographic structure [22,23]. Many scientists believe that the discovery of muscarine, or rather, its effect on the human body, was one of the most important milestones in the understanding and treatment of many diseases of the central nervous system (CNS). There are five known M_1_-M_5_ muscarinic receptors, the first three of which are stimulatory while the remaining two are inhibitory. The ailments that can be treated by understanding disorders of the murein pathway include Alzheimer’s disease, ophthalmological disorders and disorders of the respiratory and cardiovascular systems [24]. Due to the important role of this compound, many centers conduct syntheses and studies of the biological activity of muscarine analogues [25,26]. Studies show that the quaternary nitrogen atom [27], the ether oxygen atom [28] and the additional oxygen atom of the hydroxyl group are critical for the activity of muscarine. Removing the hydroxyl group resulted in a significant decrease in activity [29]. Taking these data into account, we designed and synthesized a number of derivatives containing an oxolane ring and ammonium salts of aliphatic and aromatic amines, which are analogues of muscarine, and we performed antifungal and antibacterial activity tests for them.

## 2. Results and Discussion

### 2.1. Synthesis

The commercially available 2-deoxy-D-ribose (**1**) was chosen as a substrate for obtaining the appropriate oxolane derivative. In the first stage, it was reduced to ribitol using sodium borohydride, obtaining a product with a very good yield of over 90%. The alditol thus obtained was subjected to a dehydration–cyclization reaction in a 2 molar solution of hydrogen chloride in water. In this way, after chromatographic separation, the appropriate derivative (**3**) (2*R*,3*S*)-2-(hydroxymethyl)oxolan-3-ol (**2**) was obtained (Figure 2). Although this compound is known in the literature [30], we were unable to find its NMR characterization, so we decided to include it in this work.

Compound **3** was underwent an attempted selective *O*-tosylation using tosyl chloride. Unfortunately, the use of an equimolar amount of tosyl chloride and lowering the temperature did not lead to the expected preferential tosylation of the primary hydroxyl group, and under all conditions used, a mixture of mono- and ditosyl derivatives was always formed. The resulting mixture of mono- (**4**) and ditosyl derivatives (**4′**) was separated using column chromatography to obtain the main product **4** in a moderate yield of 56%. This compound has not been previously described in the literature and its structure was confirmed by NMR, MS and IR analyses. The presence of the *O*-tosyl group only at the C-5 carbon atom (atom numbering according to Figure 3) confirms the shift of its signal in the ^13^C NMR spectrum in relation to the substrate from δ 71.89 to 69.66, while, on the contrary, the signal of the C-3 carbon atom shifted from δ 72.55 to 73.55.

The monotosyl derivative **4** was used as a substrate in the quaternization reaction with appropriate amines. In addition to trimethylamine, as in muscarine, other aliphatic and aromatic amines were used. Their structures as cations are shown in Figure 3. This choice was dictated by the fact that amines with longer hydrocarbon chains and those containing an aromatic ring have a more lipophilic character, which may facilitate their migration through the cell membrane and thus affect their biological activity.

Due to the specificity of the amines used and to obtain the highest possible yields, individual salts were obtained under different conditions, as illustrated by the data in Table 1.

Analysis of the data in Table 1 shows that, if possible, according to our previous experiences, the best yields in the quaternization reaction of derivative **4** were obtained using our previously developed [31] solvent-free method.

The structures of all obtained products were confirmed by ^1^H and ^13^C NMR spectra, in which the signals were assigned based on COSY and HETCOR spectra (all spectra are included in Appendix A). As was the case for crystals, the length of the carbon–nitrogen C-N bond was different for aliphatic and aromatic amine salts [32]. A similar pattern was observed for the chemical shifts of the terminal C-5 carbon atom associated with the positively charged nitrogen atom in the analyzed salts **5a**–**5h** (Table 2).

As shown by the data in the table above, chemical shifts of C-5 carbon atoms in aliphatic amine salts occur at higher δ values than in the case of aromatic amine salts, and among these, the aliphatic substituent on the aromatic ring also has a significant impact on this shift (compounds **5f** and **5g**).

IR spectra were also recorded for the obtained products **5a**–**5h** (Appendix A). According to the data in the literature [33], the C-N bond has strong vibration bands around 1200 cm^−1^, and the same is true of the C-O bond around 1180 cm^−1^ for the secondary alcohols of the OH group in the C-3 carbon atom.

Additionally, the structure of the obtained salts is confirmed by the recorded MALDI-TOF MS spectra. In the case of salts of organic compounds, such spectra are characterized by a strong cation signal. The situation is similar in the case of the recorded spectra of compounds **5a**–**5h** (Appendix A), where the spectrum is dominated by the characteristic signal of the ammonium salt cation (Table 3).

### 2.2. Crystal Structure and Analysis of Intermolecular Interactions

Single-crystal X-ray diffraction measurements show that compound **5e** crystallizes in orthorhombic *C*222_1_ space group with one *N*-[(2*R*,3*S*)-(3-hydroxyoxolan-2-yl)methyl]pyridinium cation and one tosylate anion in the asymmetric unit (Figure 4 and Table 4). In this space group, some pyranose and furanose derivatives also crystallize [34,35], but it is not common symmetry of crystals for these groups of compounds. In the case of muscarine iodide, this was observed in the orthorhombic space group *P*2_1_2_1_2_1_ [22], while for muscarine chloride it was observed in the space group *P*2_1_2_1_2 [23]. The length of the C6-N8 bond in compound **5e** is in good agreement with the literature data [32] regarding the length of the C-N^+^ bond in pyridinium salts and is 1.483(5) Å.

In compound **5e**, the furanose ring (O1-C2-C3-C4-C5, numbering of atoms in crystallographic descriptions consistent with the numbering of atoms in Figure 4) adopts a conformation close to the ^4^*T*_0_ (twisted on C5-O1 bond—Figure 4) [36,37] with ring-puckering parameters [38,39] θ = 0.220(8) Å and ϕ = 170(3)°, pseudorotation parameters [40] P = 259.4(16)° and τ_m_ = 25.7(6)° for the reference bond C3–C4, and delta parameter [41] Δ = 518.8°. Comparison of previously determined structures of muscarine iodide and chloride differ from each other and differ from the structure of compound **5e**. X-ray measurements of the oxolane ring of iodide [22] showed that it was almost flat. In the case of chloride [23], it had an intermediate conformation between ^3^*E* and ^3^*T*_4_, but it should be added that the numbering of atoms in the muscarine derivatives was different.

In the crystal of compound **5e**, the *N*-[(2*R*,3*S*)-(3-hydroxyoxolan-2-yl)methyl]pyridinium cation interact with tosyl anion through the O–H···O hydrogen bond to form dimer (Table 5, Figure 4). In the case of muscarine derivatives, a stabilizing hydrogen bond was found between the hydrogen atom of the hydroxyl group and the iodine or chlorine anion, respectively. Neighboring dimers are linked via C–H···O hydrogen bonds building blocks along *b*-axis (Table 5, Figure 5). Adjacent blocks are linked by C–H···O hydrogen bonds and S–O···π interactions (Table 6) [42] produced 3-D framework.

### 2.3. Microbiological Testing

Due to its low stability, compound **5f** was not included to this part of the study. Five out of six investigated derivatives, namely **5a**, **5b**, **5e**, **5g** and **5h,** exhibited neither antibacterial nor antifungal activity up to the concentration of 512 µg/mL. Compound **5c** effectively inhibited the growth of both staphylococci but only at the highest investigated concentration. The highest antifungal and antibacterial activity was exhibited in compound **5d**, with MIC values: 64 µg/mL for both staphylococci, 128 µg/mL for *E. coli* and all pathogenic yeasts strains and 512 µg/mL for *P. aeruginosa* (Table 7).

The compounds **5b**, **5c** and **5d** can be classified as surfactants with hydrophilic heads and hydrophobic tails. As mentioned above, the highest antimicrobial activity was shown by compound **5d**, which had the longest (decyl) hydrocarbon chain, while compound **5c** (octyl) exhibited only residual antibacterial potential and no activity was observed for derivative **5b** (hexyl). A similar correlation between the antimicrobial potential and the “size” of the hydrocarbon tail was also investigated in our previous study on d-xylopyranosides containing a quaternary ammonium aglycone [31]. Thus, it can be assumed that this compound (but also **5c** in the case of staphylococci), similarly to other surfactants, disrupts the integrity of the bacterial/fungal cell membrane. However, additional research is necessary to confirm this hypothesis.

The antibacterial activity of 5d against all tested strains of bacteria is significantly lower compared to gentamycin (Table 7). Much higher antibacterial activity of other classical antibiotics (in many cases with MIC values below the concentration of 1.0 µg/mL) has also been confirmed in many publications, including an article based on research carried out in our research group [43]. The antifungal activity is also lower compared to that of the most widely used antifungal agent, fluconazole (Table 7). However, the differences are not so evident as in the case of antibacterial antibiotics, e.g., MIC of fluconazole for *C. krusei* is only twice as low compared to 5d. In our opinion, the MIC values of 64 (for both staphylococci) and 128 µg/mL (for all yeast strains) do not exclude the possibility of using this compound as an antimicrobial agent (e.g., as a disinfectant). Similar or even higher values of MIC are observed for some natural products that are considered as antimicrobials, e.g., propolis [43], essential oils [44,45] and plant extracts [46,47].

## 3. Materials and Methods

### 3.1. General Section

2-Deoxy-D-ribose was purchased from Biosynth Ltd. (Compton, UK). All amines: trimethylamine, *N*,*N*-dimethylhexylamine*, N*,*N*-dimethyloctylamine, *N*,*N*-dimethyldecylamine, pyridine, 2-methylpyridine, 4-(*N*,*N*-dimethylamino)pyridine and isoquinoline were purchased from Merck (Darmstadt, Germany).

### 3.2. NMR Measurements

All measurements were carried out on a Bruker 400 MHz or 500 MHz spectrometer. ^1^H (500 or 400 MHz) and ^13^C (125 or 100 MHz, respectively) spectra were recorded in D_2_O or CDCl_3_. The signals in the spectra were assigned based on the analysis of 2D spectra (COSY and HSQC). All spectra were recorded at a controlled temperature of 298 K using a TXI inverse probe. The obtained spectra were processed and analyzed with the use of TopSpin 3.2 (Bruker BioSpin GmbH, Mannheim, Germany) software.

### 3.3. Mass Spectrometry

All samples for the MALDI-TOF mass spectrometry measurements were prepared according to the dried droplet method on a ground steel target plate with equal volumes of the sample in water and a saturated solution of α-cyano-4-hydroxycinnamic acid (CCA) in TA30 (30:70 [*v*/*v*] acetonitrile:0.1% TFA in water) or 20 mg/mL solution of 2,5-dihydroxybenzoic acid (DHB) in TA30 (30:70 [*v*/*v*] acetonitrile:0.1% TFA in water). Mass spectrometry measurements were carried out using a Bruker (Germany) AUTOFLEX MAX spectrometer and Flex Control 3.4.69 software using the reflective method in positive mode with the *m*/*z* range between 100 and 2000 and calibrated using the matrix and Bruker Peptide Calibration Standard II mass peaks. Each sample was measured with over 2500 shots and processed in Flex Analysis 3.4.79 with SNAP peak detection algorithm and signal to a threshold of 6.

### 3.4. Infrared Spestroscopy

IR spectra were recorded using an IFS66 spectrometer from BRUKER (Germany), performing Fourier transform infrared spectra with a resolution of 0.12 cm^−1^ for solid, liquid and gaseous samples in the entire range, i.e., MIR (4000 − 400 cm^−1^), FIR (700 − 4.0 cm^−1^). Spectra S53-S55 were recorded courtesy of Pro-Environment Polska Sp. z. o. o., which provided an FT-IR Spectrometer, model: Spectrum Two with ATR attachment (Spectrum Two FT-IR Spectrometer with LiTaO3 Detector, PerkinElmer, Inc., Waltham, MA, USA).

### 3.5. Polarimetry

Optical rotation was measured with a 343 PerkinElmer (Perkin Elmer, Inc., Waltham, MA, USA) polarimeter.

### 3.6. Flash Chromatography

The puriFlash 450 apparatus with a UV detector from Interchim (Montluçon, France) was used for the separation. The separation was carried out in the following solvent system: phase A—acetone (28%), phase B—hexane (72%) on a column: Puriflash Column 50 SILICA HP-Silica 50µ (40 g).

### 3.7. Melting Point Measurement

The melting point was measured using a Mel-Temp IA9000 device from Electrothermal (London, UK).

### 3.8. Single-Crystal X-ray Diffraction

Single-Crystal X-Ray Diffraction data were collected at T = 291(2) K using an Oxford Diffraction Gemini R ULTRA Ruby CCD diffractometer with MoKα (λ = 0.71073 Å) radiation (Table 1). The lattice parameters were obtained by least-squares fit to the optimized setting angles of the reflections collected by means of CrysAlis CCD and were reduced using CrysAlis RED software [48] and applying multi-scan absorption corrections. The structural resolution procedure was carried out using the SHELX [49]. The structure was solved with direct methods that carried out refinements by full-matrix least-squares on *F*^2^ using the SHELXL-2017/1 program [49]. H-atoms bound to O-atom were located on a difference Fourier map and refined freely with U_iso_(H) = 1.5U_eq_(O). H-atoms bound to C-atoms were placed geometrically and refined using a riding model with C–H = 0.93–0.97 Å and U_iso_(H) = 1.2U_eq_(C) (C–H = 0.96 Å and U_iso_(H) = 1.5U_eq_(C) for the methyl group). All interactions were found using the PLATON program [36], whereas ORTEPII [50], PLUTO-78 [51] and Mercury [52] programs were used to prepare the molecular graphics. Crystallographic data for the structure reported in this article were deposited at the Cambridge Crystallographic Data Centre, under deposition numbers No. CCDC 2325962. Copies of the data can be obtained free of charge via https://www.ccdc.cam.ac.uk/structures/ (accessed on 18 January 2024).

### 3.9. Antimicrobial Activity

Two reference strains of Gram-negative bacteria (*Escherichia coli* ATCC 25922 and *Pseudomonas aeruginosa* ATCC 27853), two reference strains of Gram-positive staphylococci (*Staphylococcus aureus* ATCC 25923 and *Staphylococcus aureus* ATCC 29213) and four reference strains of pathogenic yeasts of the genus *Candida* spp. (*Candida albicans* SC5314, *Candida glabrata* DSM 11226, *Candida krusei* DSM 6128 and *Candida parapsilosis* DSM 5784) were selected for the evaluation of the antimicrobial potential of the synthesized compounds. Determination of the MIC parameter (Minimum Inhibitory Concentration) of these substances was performed using a serial, two-fold dilution method in 96-well microtiter plates under conditions recommended by the Clinical and Laboratory Standards Institute (CLSI, Pittsburgh, PA, USA). In the case of bacterial strains, the assay was performed in autoclaved Mueller–Hinton Broth (MHB) and antifungal activity was tested in a filter-sterilized RPMI medium (RPMI—10,4 g/L; Glucose 18 g/l, MOPS—35 g/L, pH 7.0). MHB broth and all components of RPMI medium were bought from Merck (Darmstadt, Germany). Each compound was dissolved in an appropriate medium to the final concentration of 1024 µg/mL. In the next step, serial two-fold dilutions of the tested agents (over a range of concentrations from 1024.0 to 2.0 µg/mL) were prepared in the rows of 96-well microtitration plates in a final volume of 100 μL of the appropriate medium. Both yeasts and bacterial strains were cultivated on agar plates (YPD for yeasts and MHB for bacterial strains—both from Merck) for 18–24 h at 37 °C. Using a sterile loop, several colonies of each strain were harvested from the surface of agar medium and suspended in sterile PBS (phosphate-buffered saline, pH 7.4 at 25 °C, purchased from Merck) solution to obtain an optical density OD_600_ = 0.13 (for bacteria—equal to the cells concentration of approximately 1 x 10^8^ CFU/mL) and OD_660_ = 0.10 (for yeasts—equal to the cell concentration of approximately 1 x 10^6^ CFU/mL). Next, suspensions of bacterial strains were diluted in MHB broth (at a ratio of 1:100 *v*/*v*) and suspensions of yeast strains were diluted in RPMI medium (at a ratio of 1:50 *v*/*v*). Finally, 100 μL of the cells’ suspensions were loaded into the wells of 96-well microtitration plates prepared in advance, which contained 100 μL of two-fold dilutions of the tested agents. A positive growth control of each strain (growth in the medium not supplemented with any agent) as well as a negative control (medium not inoculated with bacterial/fungal cells), were included in each assay. Moreover, gentamicin (in the range of concentrations from 0.125 to 64 µg/mL) and fluconazole (in the range of concentrations (from 1 to 512 µg/mL) were used as reference antibacterial and antifungal agents, respectively. Following the incubation of the plates at 37 °C for 24 h, the determination of the MIC values of the tested agents was carried out by measuring the absorbance at 531 nm using a Victor3 microplate reader (Perkin Elmer, Inc., Waltham, MA, USA). The lowest concentration of the agent causing inhibition of growth equal to or greater than 90% (MIC90) of the growth control was taken as the MIC value. Each test was repeated in triplicate.

### 3.10. 2-Deoxy-D-Ribitol (2)

2-Deoxy-D-ribose (1.0 g, 7.455 mmol) was dissolved in water (20 mL), cooled in an ice bath and sodium borohydride (0.25 g, 6.609 mmol) was added portionwise. The reaction mixture was stirred at 5 °C. After 24 h, the mixture was acidified by adding Dowex 50WX 8-400 resin. After the resin was filtered off, the filtrate was concentrated to a thick yellow oil and dried in a vacuum desiccator over anhydrous CaCl_2_ (for 258 h). 2-Deoxy-D-ribitol (1.007 g, 7.398 mmol) was obtained in almost quantitative yield.

### 3.11. (2. R,3S)-2-(Hydroxymethyl)Oxolan-3-Ol (3)

2-Deoxy-D-ribitol (1.221 g, 8.968 mmol) was dissolved in 2M aqueous HCl (61 mL) and placed in a screw-top vessel. The solution was stirred at 100 °C for 48 h. Then, the volatile components were removed under reduced pressure and the resulting oil was dried in a vacuum desiccator over anhydrous CaCl_2_ (for 93 h). Methanol (100 mL) and activated carbon were added to the dry oil. The mixture was heated at reflux for 30 min. After this time, the mixture was filtered, and the filtrate was concentrated under reduced pressure and dried in a vacuum desiccator over CaCl_2_ for 162 h and (2*R*,3*S*)-2-(hydroxymethyl)oxolan-3-ol (**3**) was obtained (0.810 g, 6.857 mmol, 77% yield) as a colorless oil; R_f_ = 0.47 (diethyl ether–chloroform–methanol 3:2:1), [α] 37.7^o^ (*c* 1, H_2_O); ^1^H NMR (D_2_O): δ 4.22–4.19 (m, 1H, H-3), 3.95–3.87 (m, 2H, H-1, H-1′), 3.78–3.76 (m, 1H, H-4), 3.59–3.47 (m, 2H, H-5, H-5′), 2.14–2.04 (m, 1H, H-2), 1.88–1.82 (m, 1H, H-2′); ^13^C NMR (D_2_O): δ 86.17 (C-4), 72.55 (C-3), 67.14 (C-1), 71.89 (C-5), 34.13 (C-2). IR: 3375.6 cm^−1^ O-H, 1099.5 cm^−1^ C-O.

### 3.12. (2. R,3S)-2-(O-Tosylmethoxyl)Oxolan-3-Ol (4)

Anhydrous pyridine (6.6. mL, 81.937 mmol) was added to 1,4-anhydro-2-deoxy-D-ribitol (**3**) (0.444 g, 3.759 mmol). The mixture was cooled to 0 °C and *p*-toluenesulfonyl chloride (0.716 g, 3.756 mmol) was added portionwise. The reactions were carried out with stirring for another hour at 0° C and then for 24 h at room temperature. After this time, the mixture was concentrated to a thick yellow oil. TLC analysis showed the presence of mono- (**4**) and ditosyl derivatives (**4′**). The mixture was separated by column chromatography using an acetone–hexane (2:5) separation system. As a result of the separation, (2*R*,3*S*)-2-(tosylmethoxyl)oxolan-3-ol (**4**) was obtained in the form of a thick, colorless oil with a yield of 56%. After crystallization from ethanol, **4** (14%) was obtained, mp 92–93 °C and R_f_ 0.36 (acetone–hexane 2:3), [α] 28.00° (*c* 1, CHCl_3_); ^1^H NMR (CDCl_3_): δ 1.85–1.92 (m, 1H, H-2), 2.04–2.13 (m, 1H, H-2′), 2.44 (s, 3H, CH_3_Ph), 3.87–3.93 (m, 3H, H-1, H-1′, H-4), 3.99 (dd, 1H, *J*_4,5_ = 4.8, *J*_4,5′_ = 10.8, H-5), 4.08 (dd, 1H, H-5′), 4.32 (qu, 1H, *J* = 2.8, *J*_2′*,3*_ = 3.6, H-3), 7.40 and 7.78 (2d, each 2H, *J* = 7.7, Ph); ^13^C NMR (CDCl_3_): δ 145.26, 132.94, 130.13, 128.18 (C, Ph), 83.43 (C-4), 73.55 (C-3), 69.66 (C-5), 67.59 (C-1), 35.16 (C-2), 21.85 (C, PhCH_3_); MALDI TOF MS (CHCA): *m*/*z* 295.1 [M+Na]^+^). IR: 3401.3 cm^−1^ O-H, 1175.1 cm^−1^ C-O.

### 3.13. General Procedure for the Quaternization Reactions

Procedure IA

(2*R*,3*S*)-2-(*O*-tosylmethoxyl)oxolan-3-ol (**4**) was placed in a glass screw-on ampoule (volume 1.5 mL) and amine and acetonitrile were added. The reaction mixture was heated at 70 °C. The volatile components were then removed under reduced pressure. Water was added to the residue and the aqueous solution was extracted twice with chloroform to separate the product from unreacted substrate **4**. The aqueous layer (containing the product) was concentrated under reduced pressure and the product was dried at −19.7 °C over anhydrous CaCl_2_.

Procedure IB

(2*R*,3*S*)-2-(*O*-tosylmethoxyl)oxolan-3-ol (**4**) was placed in a glass screw-on ampoule (volume 1.5 mL) and amine and acetonitrile were added. The reaction mixture was heated at 70 °C. The volatile components were then removed under reduced pressure. To remove unreacted starting material **4**, diethyl ether was added to the residue and the mixture was shaken. The resulting precipitate was centrifuged and dried at −19.7 °C over anhydrous CaCl_2_.

Procedure IIA

The procedure was analogous to procedure IA but did not require the addition of acetonitrile.

Procedure IIB

The procedure was analogous to procedure IB but did not require the addition of acetonitrile.

### 3.14. N-[(2R,3S)-(3-Hydroxyoxolan-2-yl)Methyl]-N,N,N-Trimethylamonium Tosylate (5a)

A reaction of compound **4** (37.9 mg, 0.139 mmol) and a 33% methanolic solution of trimethylamine (2.99 mL) under the IIA procedure (359 h) gave the title compound **5a** as an orange oil (41.4 mg, 89%);R_f_ = 0.0 (acetone–hexane 2:3) [α] 9.9° (*c* 1, H_2_O); ^1^H NMR (D_2_O): δ 1.86–1.93 (m, 1H, H-2′), 2.13–2.21 (m, 1H, H-2) 2.34 (s, 3H, CH_3_Ph), 3.14 (s, 9H, N(CH_3_)_3_), 3.38 (dd, 1H, *J*_4,5′_ = 9.6,*J*_5,5′_ = 13.6, H-5′), 3.50 (dd, 1H, *J*_4,5_ = 1.6, H-5), 3.99 (dd, 2H, *J*_1′,2_ = 6.4, H-1′, H-1), 4.13–4.18 (m, 2H, H-3, H-4), 7.31 and 7.63 (2d, each 2H, *J* = 8.4, Ph); ^13^C NMR (D_2_O): δ 142.69, 139.86, 129.71, 125.64 (C, Ph), 79.41 (C-4), 74.46 (C-3), 67,79 (C-1), 67.62 (C-5), 54.17, 54.21, 54.24 (C, N(CH_3_)_3_), 33.14 (C, C-2), 20.73 (C, PhCH_3_); MALDI TOF MS (DHB): *m/z* 160.2 ([M-*O*Ts]^+^). IR: 3347.0 cm^−1^ O-H, 1220.0 cm^−1^ C-N, 1190.0 cm^−1^ C-O.

### 3.15. N-[(2R,3S)-(3-Hydroxyoxolan-2-Yl)Methyl]-N-Hexyl-N,N-Dimethylammonium Tosylate (5b)

**Procedure IA**: A reaction of **4** (14.6 mg, 0.054 mmol) with *N*,*N*-dimethylhexylamine (0.0186 mL, 0.107 mmol) in acetonitrile (0.5 mL) carried out for 356 h gave **5b** (11 mg, 51%); 

**Procedure IB**: A reaction of **4** (21.6 mg, 0.079 mmol) with *N*,*N*-dimethylhexylamine (0.0276 mL, 0.159 mmol) in acetonitrile (0.5 mL) carried out for 407 h gave **5b** (22.3 mg, 70%); 

**Procedure IIA**: A reaction of **4** (10.2 mg, 0.038 mmol) with *N*,*N*-dimethylhexylamine (0.013 mL, 0.075 mmol) carried out for 22 h gave **5b** (11.6 mg, 77%);

**Procedure IIB**: A reaction of **4** (23.6 mg, 0.087 mmol) with *N*,*N*-dimethylhexylamine (0.0301 mL, 0.173 mmol) carried out for 165 h gave **5b** (28.1 mg, 81%);

*R*_f_ = 0.0 (acetone–hexane 2:3); white solid; mp 76.1–76.9 °C; ^1^H NMR (CDCl_3_): δ 0.87 (t, 3H, H-f), 1.27 (s, 6H, H-c, H-d, H-e), 1.69 (s, 2H, H-b), 1.91–1.98 (m, 1H, H-2′), 2.10–2.17 (m, 1H, H-2), 2.34 (s, 3H, PhCH_3_), 3.21 and 3.22 (2×s, 6H, N(CH_3_)_2_), 3.28–4.41 (m, 3H, H-5′, NCH_2_), 3.89–3.98 (m, 2H, H-1, H-1′), 4.09–4.13 (m, 1H, H-4), 4.15–4.19 (m, 1H, *J*_3,4_ = 5.7, H-3), 4.41 (d, 1H, *J*_5,5′_ = 13.7, H-5), 7.16 and7.75 (2d, 2x2H, *J* = 8.2, Ph); ^13^C NMR (CDCl_3_): δ 142.85, 140.02, 128.86, 126.20 (C, Ph), 80.15 (C-4), 74.22 (C-3), 68.02 (C-1), 66.60 (C-5), 65.89 (C-a), 51.99–51.63 (N(CH_3_)_2_), 33.39 (C-2), 31.35 (C-d), 25.95 (C-c), 22.76 (C-b), 22.58 (C-e), 21.52 (Ph-CH_3_), 13.90 (C-f); MALDI TOF MS (CCA): m/z 230.240 ([M-*O*Ts]^+^). IR: 3392 cm^−1^ O-H, 1209 cm^−1^ C-N, 1175 cm^−1^ C-O.

### 3.16. N-[(2R,3S)-(3-Hydroxyoxolan-2-Yl)Methyl]-N,N-Dimethyl-N-Octylammonium Tosylate (5c)

**Procedure IA**: A reaction of **4** (17.0 mg, 0.062 mmol) with *N*,*N*-dimethyloctylamine (0.0257 mL, 0.125 mmol) in acetonitrile (0.5 mL) carried out for 117 h gave **5c** (11.8 mg, 44%);

**Procedure IB**: A reaction of **4** (25.5 mg, 0.094 mmol) with *N*,*N*-dimethyloctylamine (0.0385 mL, 0.187 mmol) in acetonitrile (0.5 mL) carried out for 407 h gave **5c** (32.4 mg, 81%);

**Procedure IIB**: A reaction of **4** (24.4 mg, 0.090 mmol) with *N*,*N*-dimethyloctylamine (0.0369 mL, 0.180 mmol) carried out for 165 h gave **5c** (30.7 mg, 80%);

*R*_f_ = 0.0 (acetone–hexane 2:3); white solid; mp 54.1–54.4 °C; ^1^H NMR (CDCl_3_): δ 0.88 (t, 3H, H-h), 1.26 (*b*s, 10H, H-c, H-d, H-e, H-f, H-g), 1.63–1.72 (m, 2H, H-b), 1.91–1.97 (m, 1H, H-2′), 2.10–2.17 (m, 1H, H-2), 2.34 (s, 3H, PhCH_3_), 3.21 and 3.23 (2×s, 6H, N(CH_3_)_2_), 3.28–3.41 (m, 3H, H-5′, NCH_2_), 3.89–3.98 (m, 2H, H-1, H-1′), 4.09–4.13 (m, 1H, *J*_4,5′_ = 5.4, H-4), 4.14–4.18 (m, 1H, *J*_3,4_ = 5.6, H-3), 4.42 (d, 1H, *J*_5,5′_ = 13.6, H-5), 7.15 and 7.75 (2d, 2x2H, *J* = 8.1, Ph); ^13^C NMR (CDCl_3_): δ 143.25, 139.70, 128.86, 125.95 (C, Ph), 80.23 (C-4), 74.31 (C-3), 67.94 (C-1), 66.48 (C-5), 65.84 (C-a), 51.73–51.36 (N(CH_3_)_2_), 33.51 (C-2), 31.60 (C-f), 29.07 (C-d, C-e),) 26.23 (C-e), 22.79 (C-b), 22.63 (C-g), 21.40 (Ph-CH_3_), 14.21 (C-h); MALDI TOF MS (CCA): m/z 258.240 ([M-*O*Ts]^+^). IR: 3342 cm^−1^ O-H, 1222 cm^−1^ C-N, 1173 cm^−1^ C-O.

### 3.17. N-[(2R,3S)-(3-hydroxyoxolan-2-yl)methyl]-N-decyl-N,N-dimethylammonium tosylate (5d)

**Procedure IB**: A reaction of **4** (24.5 mg, 0.090 mmol) with *N*,*N*-dimethyldecylamine (0.0429 mL, 0.180 mmol) in acetonitrile (0.5 mL) carried out for 331 h gave **5d** (30.8 mg, 75%); 

**Procedure IIB**: A reaction of **4** (26.0 mg, 0.096 mmol) with *N*,*N*-dimethyldecylamine (0.0455 mL, 0.191 mmol) carried out for 165 h gave **5d** (32.6 mg, 75%);

*R*_f_ = 0.0 (acetone–hexane 2:3); white solid; mp 60.3–61.2 °C; ^1^H NMR (CDCl_3_): δ 0.88 (t, 3H, H-j), 1.24 (*b*s, 14H, H-c, H-d, H-e, H-f, H-g, H-h, H-i), 1.62–1.74 (m, 2H, H-b), 1.91–1.97 (m, 1H, H-2′), 2.10–2.17 (m, 1H, H-2), 2.34 (s, 3H, PhCH_3_), 3.22 and 3.23 (2×s, 6H, N(CH_3_)_2_), 3.28–3.41 (m, 3H, H-5′, NCH_2_), 3.89–3.98 (m, 2H, H-1, H-1′), 4.09–4.13 (m, 1H, *J*_4,5′_ = 10.0, H-4), 4.15–4.18 (m, 1H, H-3), 4.43 (d, 1H, *J*_5,5′_ = 13.6, H-5), 7.15–7.75 (2d, 2x2H, *J* = 8.1, Ph); ^13^C NMR (CDCl_3_): δ 143.18, 139.63, 128.70, 125.95 (C, Ph), 80.20 (C-4), 74.26 (C-3), 67.86 (C-1), 66.62 (C-5), 65.91 (C-a), 51.34–51.78 (N(CH_3_)_2_), 33.32 (C-2), 31.87 (C-h), 29.44 (C-d), 29.37 (C-e), 29.24 (C-f), 29.10 (C-g), 26.21(C-c), 22.74 (C-b), 22.67 (C-i), 21.31 (Ph-CH_3_), 14.12 (C-j); MALDI TOF MS (CCA): m/z 286.262 ([M-*O*Ts]^+^). IR: 3336 cm^−1^ O-H, 1220 cm^−1^ C-N, 1184 cm^−1^ C-O.

### 3.18. N-[(2R,3S)-(3-hydroxyoxolan-2-yl)methyl]pyridinium tosylate (5e)

**Procedure IIA**: A reaction of **4** (13.7 mg, 0.050 mmol) with anhydrous pyridine (0.280 mL, 3.452 mmol) carried out for 296 h gave **5e** (76.5 mg, 86%); *R*_f_ = 0.0 (acetone–hexane 2:3); yellow crystals; mp 111.5–113.8 °C;); [α] 24.6^o^ (*c* 1, H_2_O); ^1^H NMR (D_2_O): δ 1.91–1.98 (m, 1H, *J*_2′,3_ = 3.2, H-2′), 2.07–2.16 (m, 1H, H-2), 2.32 (s, 3H, PhCH_3_), 3.95 (dd, 1H, *J*_1′,2_ = 7.2, H-1′), 3.99–4.05 (m, 1H, H-1), 4.17 (dt, 1H, *J*_4,5_ = 3.2, H-4), 4.34 (qvint, 1H, *J*_3,4_ = 6.0, H-3), 4.44 (dd, 1H, *J*_4,5′_ = 9.6, H-5′), 4.80 (dd, 1H, *J*_5,5′_ = 13.6, H-5), 7.29 and 7.61 (2d, each 2H, *J* = 8.4, Ph), 8.02, 8.52, 8.76 (5H, Py); ^13^C NMR (D_2_O): δ 146.41, 144,94, 128.38 (C, Py), 142.60, 139.70, 129.66, 125.60 (C, Ph), 83.97 (C-4), 73.10 (C-3), 67.77 (C-1), 62.84 (C-5), 33.66 (C-2), 20.69 (C, PhCH_3_); MALDI TOF MS (DHB): *m/z* 180.2 ([M-*O*Ts]^+^). IR: 3331.0 cm^−1^ O-H, 1215.5 cm^−1^ C-N, 1175.8 cm^−1^ C-O.

### 3.19. N-[(2R,3S)-(3-hydroxyoxolan-2-yl)methyl]-2-methylpyridinium tosylate (5f)

**Procedure IIA**: A reaction of **4** (26.9 mg, 0.099 mmol) with 2-methylpyridine (0.620 mL, 6.258 mmol) carried out for 496 h gave **5f** (37.7 mg, 75%); compound unstable at room temperature, *R*_f_ = 0.0 (acetone—hexane 2:3); yellowish oil; ^1^H NMR (CDCl_3_): δ 1.93–1.99 (m, 1H, H-2′), 2.11–2.18 (m, 1H, H-2), 2.33 (s, 3H, PhCH_3_), 2.89 (s, 3H, CH_3_Py) 3.86–3.89 (m, 2H, H-1, H-1′), 4.14 (*b*s, 1H, H-4), 4.38 (*b*s, 1H, H-3), 4.59–4.63 (m, 1H, H-5′), 5.36 (d, 1H, *J*_5,5′_ = 13.9, H-5), 7.14 (d, 2H, Ph), 7.68–7.77 (m, 4H, Ph, H-b, H-d/Py), 8.18 (t, 1H, H-c/Py), 9.22 (d, 1H, *J* = 6.0, H-a/Py); ^13^C NMR (CDCl_3_): δ 155.94–144.55, 128.87 and 125.96 (C, Py), 139.78, 129.44 and 125.56 (C, Ph), 85.24 (C-4), 73.12 (C-3), 67.87 (C-1), 59.87 (C-5), 34.10 (C-2), 21.25 (2C, PhCH_3_ and PyCH_3_); MALDI TOF MS (DHB): *m/z* 194.095 ([M-*O*Ts]^+^). IR: 3339.7 cm^−1^ O-H, 1218.1 cm^−1^ C-N, 1189.0 cm^−1^ C-O.

### 3.20. N-[(2R,3S)-(3-hydroxyoxolan-2-yl)methyl]-4-(N,N-dimethylamino)pyridinium tosylate (5g)

**Procedure IIA**: A reaction of **4** (25.7 mg, 0.094 mmol) with 4-(*N*,*N*-dimethylamine)pyridine (15.9 mg, 0.130 mmol) carried out for 166 h gave **5g** (43.9 mg, 79%); *R*_f_ = 0.0 (acetone—hexane 2:3); colorless oil; ^1^H NMR (CDCl_3_): δ 1.91–1.97 (m, 1H, H-2′), 1.98–2.04 (m, 1H, H-2), 2.33 (s, 3H, PhCH_3_), 3.20 (s, 6H, N(CH_3_)_2_), 3.82–3.90 (m, 2H, H-1, H-1′), 4.07 (*b*s, 1H, H-4), 4.20 (*b*s, 2H, H-3, H-5′), 4.79 (d, 1H, *J*_5,5′_ = 13.3, H-5), 7.15 and 7.79 (2d, each 2H, *J* = 9.0, Ph), 6.76 and 8.41 (2d, 4H, *J* = 7.0, Py); ^13^C NMR (CDCl_3_): δ 156.23–143.37, 107.50 (5C, Py), 142.60–126.02 (6C, Ph), 84.99 (C-4), 72.12 (C-3), 67.54 (C-1), 59.26 (C-5), 40.25 (2C, N(CH_3_)_2_), 34.21 (C-2), 21.34 (C, PhCH_3_); MALDI TOF MS (CCA): *m/z* 223.167 ([M-*O*Ts]^+^). IR: 3349.2 cm^−1^ O-H, 1205.6 cm^−1^ C-N, 1181.7 cm^−1^ C-O.

### 3.21. N-[(2R,3S)-(3-hydroxyoxolan-2-yl)methyl]isoquinolinium tosylate (5h)

**Procedure IIA**: A reaction of **4** (28.5 mg, 0.105 mmol) with isoquinoline (0.287 g, 2.222 mmol) carried out for 306 h gave **5h** (27.1 mg, 64%); *R*_f_ = 0.0 (acetone—hexane 2:3); colorless oil; ^1^H NMR (CDCl_3_): δ 1.89–1.96 (m, 1H, H-2′), 2.05–2.12 (m, 1H, H-2), 2.28 (s, 3H, PhCH_3_), 3.87–3.92 (m, 2H, H-1, H-1′), 4.28 (*b*s, 1H, H-4), 4.34 (*b*s, 1H, H-3), 4.79–4.83 (m, 1H, H-5′), 5.49 (d, 1H, *J*_5,5′_ = 16.0, H-5), 7.10 and 7.77 (2d, each 2H, *J* = 8.0, Ph), 7.86–10.28 (7H, *J* = 6.8, amine); ^13^C NMR (CDCl_3_): δ 150.93–131.05 and, 127.76–126.90 (C, Ph-amine), 142.81, 128.82–125.59 (C, Ph), 85.24 (C-4), 72.86 (C-3), 67.78 (C-1), 63.00 (C-5), 34.15 (C-2), 21.38 (C, PhCH_3_); MALDI TOF MS (CCA): *m/z* 230,130 ([M-*O*Ts]^+^). IR: 3344.9 cm^−1^ O-H, 1215.9 cm^−1^ C-N, 1186.6 cm^−1^ C-O.

## 4. Conclusions

2-Deoxy-D-ribose is a commercially available compound and, as the performed syntheses have shown, it can be successfully used for the synthesis of muscarine-type derivatives. Eight new derivatives were synthesized, which are analogues of muscarine (containing an oxolane ring with a hydroxyl group and a quaternary nitrogen atom attached to it). NMR (1D and 2D), MS and IR analyses fully confirmed their purity and structure. The X-ray measurements confirmed that the pyridinium derivative crystallizes in the orthorhombic space group, similarly to muscarine derivatives. The oxolane ring assumes a twisted *T* conformation, as in muscarine iodide. The microbiological tests carried out showed that the highest activity was demonstrated by derivatives containing an eight- and ten-carbon chain at the quaternary nitrogen atom, respectively. This is most likely the result of their strongest hydrophobic character, which increases their affinity for the hydrophobic cell wall of bacteria or fungi, but this requires additional and extended research.

## Figures and Tables

**Figure 1 ijms-25-02368-f001:**
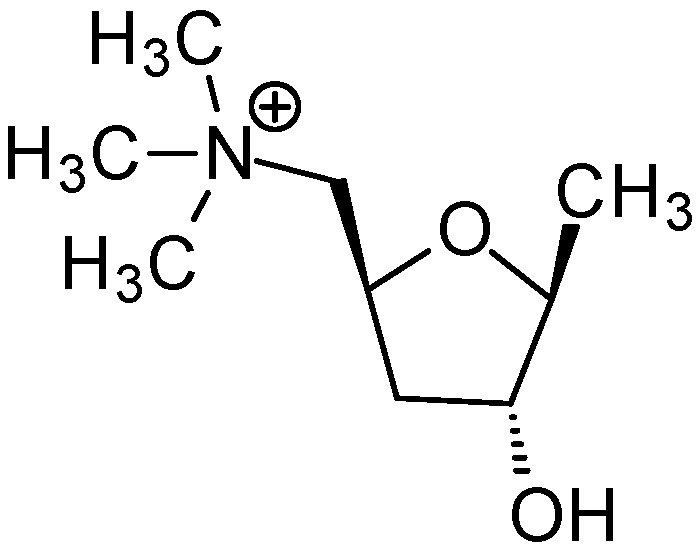
Muscarine [(2*S*,4*R*,5*S*)-4-hydroxy-5-methyloxolan-2-yl]methyl-trimethylazanium.

**Figure 2 ijms-25-02368-f002:**
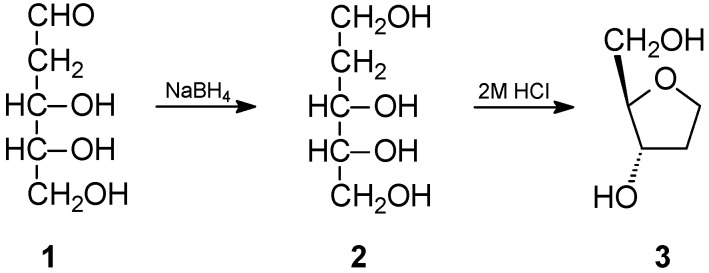
Scheme of obtaining the oxolane derivative (**3**) from 2-deoxy-D-ribose.

**Figure 3 ijms-25-02368-f003:**
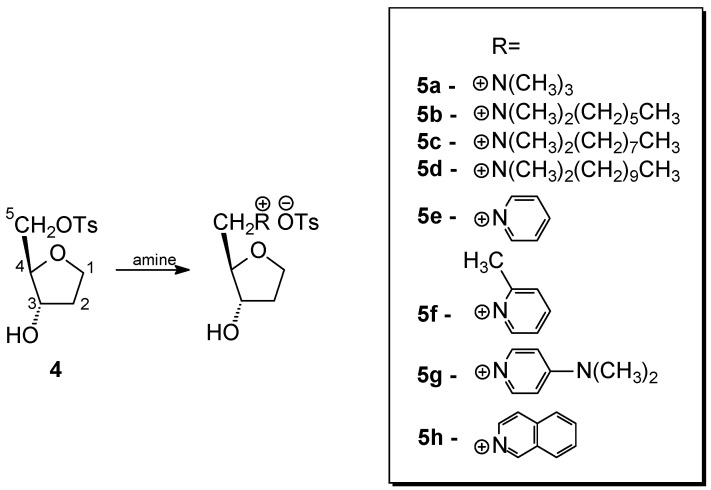
Scheme of the synthesis of quaternary ammonium salts of oxolane derivative.

**Figure 4 ijms-25-02368-f004:**
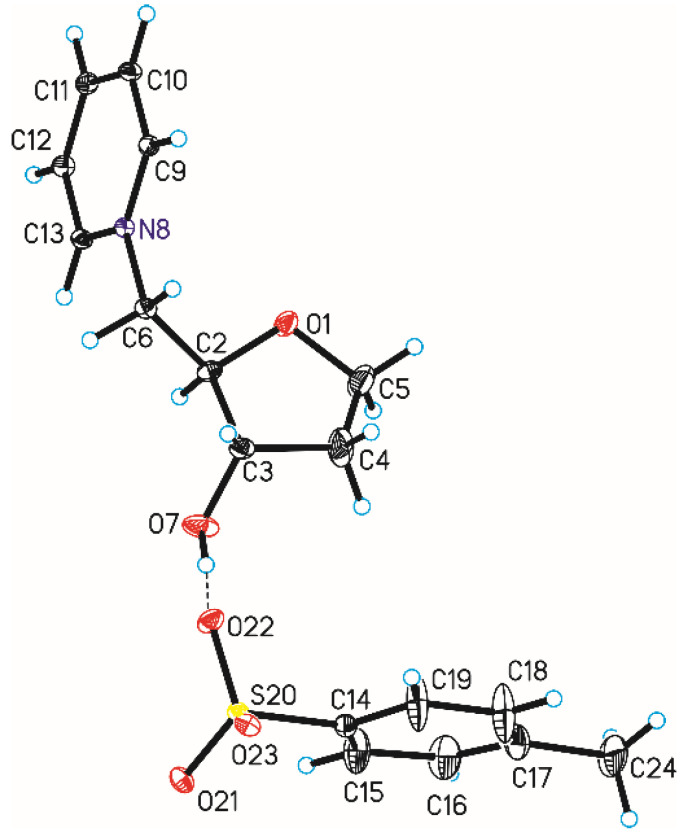
Molecular structure of compound **5e**, showing the atom-labeling scheme. Displacement ellipsoids are drawn at the 25% probability level and H atoms are shown as small spheres of arbitrary radius (hydrogen bond are represented by dashed line).

**Figure 5 ijms-25-02368-f005:**
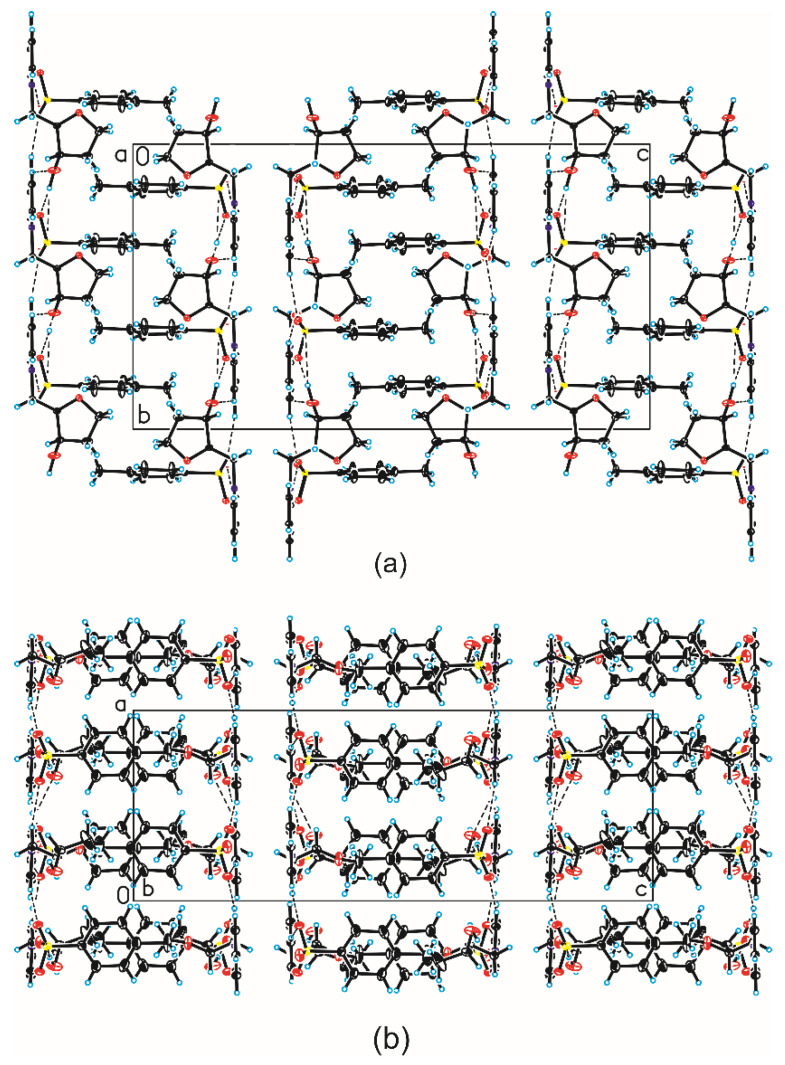
Crystal packing of compound **5e**: (**a**) viewed along the *a*-axis and (**b**) viewed along the *b*-axis (hydrogen bonds are represented by dashed lines).

**Table 1 ijms-25-02368-t001:** Appropriate reaction yields [%] for obtaining ammonium salts **5a**–**5h** in the synthesis procedures used.

Product	Procedure IA	Procedure IB	Procedure IIA	Procedure IIB
	With Solvent		No Solvent	
**5a**	-	-	89	-
**5b**	51	70	77	81
**5c**	44	81	-	80
**5d**	-	75	-	75
**5e**	-	-	86	-
**5f**	-	-	75	-
**5g**	-	-	79	-
**5h**	-	-	64	-

**Table 2 ijms-25-02368-t002:** Chemical shifts of the C-5 carbon atom in compounds **5a**–**5h**. Shaded areas contain salts of aromatic amines.

Compound	5a	5b	5c	5d	5e	5f	5g	5h
δ [ppm]	67.62	66.60	66.48	66.62	62.84	59.87	59.26	63.00

**Table 3 ijms-25-02368-t003:** M/z values of dominant ions in MALDI-TOF MS spectra [M-OTs] of compounds **5a**-**5h**.

Compound	5a	5b	5c	5d	5e	5f	5g	5h
Molecular mass	331.34	401.38	429.38	457.40	351.42	365.23	394.31	401.27
*m*/*z*	160.20	230.24	258.24	286.26	180.20	194.09	223.17	230.13

**Table 4 ijms-25-02368-t004:** Crystal data and structure refinement parameters for compound **5e**.

Compound	5e
Chemical formula	C_10_H_14_O_2_N^+^ ∙ C_7_H_7_SO_3_^−^
FW/g·mol^−1^	351.41
Crystal system	Orthorhombic
Space group	*C*222_1_
*a*/Å	9.645 (2)
*b*/Å	14.500 (3)
*c*/Å	26.270 (5)
*α*/°	90
*β*/°	90
*γ*/°	90
*V*/Å^3^	3673.9 (13)
*Z*	8
*T*/K	291 (2)
*λ*_Mo_/Å	0.71073
*ρ_cal_*_c_/g·cm^–3^	1.271
*F(000)*	1488
µ/mm^−1^	0.201
*θ* range/°	3.102–25.002
Completeness *θ*/%	99.5
Reflections collected	6422
Reflections unique	3178 [R_int_ = 0.0388]
Data/restraints/parameters	3178/0/221
Goodness of fit on *F*^2^	0.864
Final R_1_ value (*I* > 2σ(*I*))	0.0509
Final *w*R_2_ value (*I* > 2σ(*I*))	0.1084
Final R_1_ value (all data)	0.1130
Final *w*R_2_ value (all data)	0.1211
Absolute structure parameter	−0.07 (10)
CCDC number	2325962

**Table 5 ijms-25-02368-t005:** Hydrogen bonding geometry for compound **5e**.

D–H···A	*d*(D-H) [Å]	*d*(H···A) [Å]	*d*(D⋯A) [Å]	∠D–H⋯A (°)
O7–H7A···O22	1.04 (9)	1.66 (9)	2.696 (6)	177 (12)
C6–H6B···O21 ^i^	0.97	2.41	3.328 (7)	158
C9–H9A···O21 ^i^	0.93	2.44	3.293 (7)	152
C10–H10A···O7 ^i^	0.93	2.56	3.293 (9)	136
C11–H11A···O21 ^ii^	0.93	2.51	3.428 (7)	169
C13–H13A···O23 ^iii^	0.93	2.34	3.257 (7)	168
Symmetry code: (i) −1/2 + x,1/2 + y,z; (ii) x,1 + y,z; (iii) 1/2 + x,1/2 + y,z.				

**Table 6 ijms-25-02368-t006:** S–O···π interactions geometry for compound **5e**.

S–O···Cg *	*d*(O···Cg) [Å]	*d*(S⋯Cg) [Å]	∠S–O⋯Cg [°]
S20–O22···Cg ^iv^	3.325 (5)	4.168 (3)	116.3 (2)
Symmetry code: (iv) 1/2 − x,−1/2 + y,1/2 − z.			

* Cg denote the N8/C9/C10/C11/C12/C13 ring centroid.

**Table 7 ijms-25-02368-t007:** Antimicrobial activity of the synthesized compounds.

Comp.	MIC_90_ Values [µg/mL]							
	Gram-Negative Bacteria		Gram-Positive Bacteria		Pathogenic Yeasts from the Genus *Candida* spp.			
	*E. coli* ATCC 25922	*P. aeruginosa* ATCC 27853	*S. aureus* ATCC 25923	*S. aureus* ATCC 29213	*C. albicans* SC5314	*C. glabrata* DSM 11226	*C. krusei* DSM 6128	C. *parapsilosis* DSM 5784
**5a**	>512	>512	>512	>512	>512	>512	>512	>512
**5b**	>512	>512	>512	>512	>512	>512	>512	>512
**5c**	>512	>512	512	512	>512	>512	>512	>512
**5d**	**128**	**512**	**64**	**64**	**128**	**128**	**128**	**128**
**5e**	>512	>512	>512	>512	>512	>512	>512	>512
**5f**	ND *	ND *	ND *	ND *	ND *	ND *	ND *	ND *
**5g**	>512	>512	>512	>512	>512	>512	>512	>512
**5h**	>512	>512	>512	>512	>512	>512	>512	>512
**Fluconazole**	ND	ND	ND	ND	1	16	64	2
**Gentamicin**	1	0.5	0.125	0.125	ND	ND	ND	ND

ND *—antimicrobial activity was not determined because of low stability of this compound.

## Data Availability

Data are contained within the article and Appendix A.

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
