# Peer review of "Oxolane Ammonium Salts (Muscarine-Like)—Synthesis and Microbiological Activity"

_ijms, 2024, doi:10.3390/ijms25042368_

Round 1

Reviewer 1 Report

Comments and Suggestions for Authors

The manuscript by Bogdanowicz  P. et al describes the synthesis and characterization of some new oxolane ammonium salts, as muscarine derivatives, and their antimicrobial activity evaluation on some  reference strains of Gram-negative bacteria (Escherichia coli ATCC 25922 and Pseudomonas aeruginosa ATCC 27853), of Gram-positive staphylococci (Staphylococcus aureus 17 ATCC 25923 and Staphylococcus aureus ATCC 29213) and four reference strains of pathogenic yeasts of the genus Candida spp. (Candida albicans SC5314, Candida glabrata DSM 11226, Candida krusei DSM 19 6128 and Candida parapsilosis DSM 5784).  
Overall, the presented research and results are important and interesting for the readership,
but there are some inadvertences that the authors should solve before this paper to be accepted for publication. Also, there are many typographical errors.

 - Please indicate which of the compounds are new, uncited in the literature.

In the case the compounds known in the literature, please add the reference for the protocol used to obtain of these.

- Although the melting point was determined for some of the compounds, in section 3, the authors did not indicate the apparatus used. Also, in this section, the protocol for recording the mass and NMR spectra should be fully described. At 3.2., line 188, the authors indicated that ’’All measurements were carried out on a Bruker 400 MHz or 500 MHz spectrometer’’ but the 13C-NMR were carried out at 100 or 120 MHz, according to supplementary material.

What solvents were used to record the spectra of compounds 5, D2O or CDCl3? There are many inconsistencies between the spectral data presented in section 3 and the spectra presented in the supplementary material. For example, for 5b, in the section 3.14, it was written that the NMR spectra were registered in CDCl3, while in the supplementary material, in the caption of the 1H- and 13C-NMR spectra, appear D2O, but CDCl3 on the spectra (see pictures). The same errors for almost all compounds.

Please check the NMR spectral characterization data of the synthesized compounds; in some cases, some protons or carbon atoms are missing. e.g. for 4, the protons CH3, OH; some carbon atoms from 5c, 5d, 5f.

- in section 3, at the characterization of the compounds, should be added the most important IR absorption bands

- the abstract and conclusions section should be improved. If the compounds are new, it should be indicated that their structure has been confirmed by NMR, IR spectra in addition to X-rays.

- Line 73-77, lines 92-94, please check. In the results and discussion section,  the confirmation of the structure synthesized compounds should be improved.

 - figure 3, R fragment from 5c should be +N(CH3)2(CH2)7CH3 and not  +N(CH3)2(CH3)7CH3

 The results of antimicrobial evaluation studies of the synthesized compounds are weak. For comparison. However, a reference should be used.

Line 178, compound 5b or 5c?

Comments on the Quality of English Language

Minor editing of English language required. 

Reviewer 2 Report

Comments and Suggestions for Authors

This note concerns the synthesis and structural identification of a series of 8 tosylate cationic oxolanic derivatives closed to muscarine. It’s known that (+)-Muscarine tosylate is an agonist of prototype mAChR and is a toxin that can stimulate the parasympathetic nervous system. This work is a continuation of the team's research activities of glycocationic glycosides. The structure of the new analogues was confirmed using NMR spectroscopy (1H, 13C, COSY and HETCOR), single-crystal X-ray diffraction measurements that confirmed that the pyridinium derivative crystallizes in the orthorhombic space group. The antimicrobial activity of the synthesized compounds was determined. The bibliographical references are judicious but could have been a little more up to date. The writing of the article needs to be reworked before publication.

Line 73 : « The presence of the O-tosyl group only at the C-6 carbon atom (atom numbering according to Fig.4) ». For greater clarity, please indicate the carbon numbering in Figure 4, as shown in the text.

Line 86 : « Due to the specificity of the amines used and to obtain the highest possible yields, individual salts were obtained under different conditions, as illustrated by the data in Table 1. » : Please give details of the synthesis: the reactions are not optimized. Reading the experimental protocols, reaction times are extremely long (several days) and reactions are carried out on very small quantities.

Line 160 : For biological evaluation, the presence of reference antimicrobial molecules would be a plus for comparing the efficacy of synthesized molecules.

spelling mistakes : Line 184 : N,N-dimethylohexyloamine, N,N-dimethylookoctylamine, N,N-dimethylodecyloamine ; line 209 : Puriflesh Column 50

Line 264 : « 2-Deoxy-D-ribose (1.0 g, 7.455 mmol) was dissolved …. 2-Deoxy-D-ribitol (1.315 g, 9.658 mmol) was obtained in almost quantitative yield. » Poorly formulated protocol. You can't get more mol than the starting compound. Please write: quantitative crude yield, and dry the compound longer.

In the experimental part : Review the description of certain NMR spectra. Some multiplets are described by a single chemical shift value instead of two values to frame the bulk.

References 34 and 43 are identical. Please correct and standardize the writing of the references. For example reference 20 is in italics.

Figure 1 and Figure 2 : Review the drawings of the molecules to agree with the indicated stereochemistry

Round 2

Reviewer 1 Report

Comments and Suggestions for Authors

 The authors have revised the article, but it still has some omissions or errors.

 - lines 79-82: please, check! the value 61.89 ppm, doesn't seem to exist; should be 71.89 ppm; on the other hand, according to 3.11 and 3.12 section, the value of 86.71 to 83.43 ppm belong to C-4, not C-3.

- according to line 326, the 13C-NMR was registered in D2O (on the other hand, for 1H-NMR was used CDCl3), but in the supplementary material appears in CDCl3. Please, check again!

 - similar to other new compounds, in the supplementary material please add, at the NMR spectra figure, the structure of compounds 5a, 5e, 5g, 5h.

- in the case of protocols used for the synthesis of compounds 2 and 3 known in the literature, the authors should add the bibliographic references

- also, the author should add the protocol for the mass spectra

- line 77, ’’the main product 4 in a good yield of 56%.’’ I suggest to add a moderate instead of good

- those two references from 22 (a and b) should be written separately.

Reviewer 2 Report

Comments and Suggestions for Authors

The authors have made most of the proposed corrections.

But it seems that the changes to the NMR spectra are not complete, many multiplets are still described with a single value and not 2, they have not been changed in the new version. For example line. Please review the description of all compounds. For example  Lines 323-324 : 1H NMR (CDCl3): d 1.88 (m, 1H, H-2), 2.09 (m, 1H, H-2’), 2.44 (s, 3H, CH3Ph), 3.89 (m, 3H, H-1, H-1’, H-4), 3.99 (dd, 1H,

The authors : « We would like to thank the reviewer for drawing our attention to the shortcomings in the description of the NMR spectra. We carefully analysed the spectra again and in the revised text we made the description more consistent, corrected errors and included missing parameters. »
